# Peer Influence during Adolescence: The Moderating Role of Parental Support

**DOI:** 10.3390/children8040306

**Published:** 2021-04-17

**Authors:** Mazneen Havewala, Julie C. Bowker, Kelly A. Smith, Linda Rose-Krasnor, Cathryn Booth-LaForce, Brett Laursen, Julia W. Felton, Kenneth H. Rubin

**Affiliations:** 1Department of Counseling, Higher Education, and Special Education, University of Maryland, College Park, MD 20742, USA; 2Department of Psychology, University of Buffalo, Buffalo, NY 14260, USA; jcbowker@buffalo.edu; 3Department of Human Development and Quantitative Methodology, University of Maryland, College Park, MD 20742, USA; ksmith56@umd.edu (K.A.S.); krubin@umd.edu (K.H.R.); 4Department of Human Psychology, Brock University, St. Catharines, ON L2S 3A1, Canada; linda.rose-krasnor@brocku.ca; 5Center on Human Development and Disability, University of Washington, Seattle, WA 98195, USA; ibcb@uw.edu; 6Department of Psychology, Florida Atlantic University, Fort Lauderdale, FL 33314, USA; laursen@fau.edu; 7Center for Health Policy & Health Services Research, Henry Ford Health Systems, Detroit, MI 48202, USA; jfelton4@hfhs.org

**Keywords:** peer influence, parental support, internalizing problems, externalizing problems, adolescence

## Abstract

Although many studies show that peers influence the development of adolescent internalizing and externalizing difficulties, few have considered both internalizing and externalizing difficulties in the same study, and fewer have considered the contributions of parents. Using a longitudinal sample of 385 adolescents, the contributions of best friends’ internalizing and externalizing difficulties (as assessed in Grade 6; G6: *M*_age_ = 13.64 years; 53% female; 40% ethnic or racial minority) were examined as they predicted subsequent adolescent internalizing and externalizing difficulties (at G8); in addition, the moderating role of both maternal and paternal support (at G6) was explored. Structural equation modelling revealed that best friend internalizing difficulties predicted decreases, but that best friend externalizing difficulties predicted increases in adolescents’ externalizing difficulties over time. Significant interactions involving both maternal and paternal support revealed that the negative impact of a G6 best friend having internalizing problems on later G8 adolescent externalizing problems was stronger at low levels of maternal and paternal support. The findings highlight the complex, and interactive, influences of friends and parents on the development of internalizing and externalizing symptomatology during adolescence, and underscore the importance of targeting both sources of social influence in research and clinical work.

## 1. Introduction

Due to its numerous developmental changes and stressors, adolescence is considered a vulnerable and risky developmental period during which internalizing (i.e., anxiety, depressive symptoms) and externalizing (i.e., rule-breaking, delinquency) problems often first emerge, become stable, and result in impairment in other domains [1]. For instance, a rise in internalizing and externalizing difficulties as children transition into adolescence has been documented [2]. Clear findings also show that internalizing and externalizing difficulties during adolescence interfere with positive functioning with peers and academic achievement, and that externalizing difficulties confer risk for substance use and abuse, violent behavior, and crime [3,4]. Thus, identifying risk as well as protective factors associated with the development and maintenance of both internalizing and externalizing difficulties is an important endeavor.

There is robust evidence linking peers to the development of internalizing and externalizing difficulties during adolescence via peer contagion processes. In general, adolescents who have friends with internalizing or externalizing problems are more likely to select, but also become similar to their friends over time [5,6]. It is rare for studies in this area, however, to evaluate both internalizing and externalizing problems simultaneously, both in terms of the influence of friends and adolescent outcomes. This is significant given the strong associations between internalizing and externalizing difficulties during childhood and adolescence, and evidence suggesting the potential cross-over of difficulties (i.e., internalizing difficulties might lead to externalizing difficulties and vice versa) [7]. Given the potential clinical implications of understanding the influence of peers on the development of internalizing and externalizing problems and the development of intervention modules targeted towards diminishment of internalizing or externalizing difficulties, it is important to understand the unique effects of a friend’s internalizing and externalizing difficulties on the development of both adolescent internalizing and externalizing problems.

The focus on peer influence on the development of internalizing and externalizing difficulties has also often been accompanied by the neglect of parent–child relationships as moderators or mediators of these difficulties. Researchers have shown direct effects of maternal support on indices of adolescent well-being, such as lower rates of internalizing and externalizing problems [8,9]. There is also evidence revealing that the quality of maternal–child relationships can protect adolescents from the effects of stress on psychopathology [10]. Research is limited, however, on the potential moderating effects of maternal support on the links between friend internalizing and externalizing difficulties and later adolescent internalizing and externalizing difficulties. The extant literature on paternal support and youth mental health is especially sparse, and no studies, to the best of our knowledge, have examined paternal support as a moderator of peer contagion. Thus, the aims of the current study were to examine the unique effects of best friend internalizing and externalizing symptoms as risk factors, and the protective role of maternal and paternal support, in the development of adolescent internalizing and externalizing symptoms.

### 1.1. Peer Contagion and Youth Internalizing and Externalizing Symptoms

There is an ever-growing body of research showing that adolescent friendships can provide a unique context for the contagion of mental health difficulties and psychopathology [11]. That is, adolescents appear to be at increased risk for the development of adjustment difficulties when their friends are experiencing difficulties themselves. For example, Stevens and Prinstein [12] found that among young adolescents in Grades 6 through 8, best friends’ level of depressive symptoms was associated with adolescents’ own depressive symptoms at follow up (11 months after the initial assessment). Similar contagion effects are well-documented for a wide variety of externalizing behaviors, including relational and instrumental aggression [13], alcohol and drug use [14,15], risky sexual behavior [6], and suicidal behavior [16].

Different theories have been suggested to explain the mechanisms of peer contagion. For example, co-rumination, or the tendency of friends to engage in excessive problem talk dwelling on negative affect [17], is one such proposed mechanism of depression contagion [18]. Co-rumination is common in adolescent friendships and may reinforce depressive/negative thoughts and beliefs, which in turn may lead to increases in both friends’ depressive symptoms. Co-rumination might also help to explain why friends become more similar in anxiety over time.

Another possible mechanism of peer contagion is deviancy training, which can be described as a process by which peers influence each other in deviant attitudes, talk, and behavior by encouraging and positively reinforcing aberrant actions [19]. Findings from several studies support the role of deviancy training in the development and maintenance of such externalizing difficulties as criminal and violent behavior, arrests, drug use, delinquency, and sexual promiscuity [20,21].

While there is extensive evidence of peer contagion effects within multiple domains of internalizing and externalizing difficulties, there have been few studies in which the effects of peer/friend maladaptive behavior of one type (e.g., externalizing symptoms) predicting youth maladaptation of another type (e.g., internalizing symptoms) have been examined (referred to hereafter as cross-over effects). Findings from the parenting literature, however, lend credence to this possibility: mothers of children with ADHD, for example, experience greater distress [22] and are more likely to experience depressive symptoms [23] compared to mothers of typically developing children. Likewise, parent alcohol use (falling under the broader category of externalizing behaviors) is a predictor of depression (a type of internalizing behavior) in young adults [24]. Thus, it is possible that being with a friend with one type of mental health issue could impact youth problems in another domain via increased general stress or the peer contagion processes described above.

Raposa et al. [25] found that best friend’s general psychopathology (a composite of internalizing symptoms, externalizing symptoms and personality psychopathology) predicted the development of adolescents’ own depressive symptoms over time. While this study was one of the first to demonstrate the broader effects of best friend’s psychopathology on youth depressive symptoms, research on the cross-over effects of internalizing and externalizing behaviors is generally absent, which not only raises questions about cross-over effects but also about unique effects of best friend internalizing and externalizing difficulties on the development of adolescent internalizing and externalizing difficulties. This latter point is significant given the strong associations between internalizing and externalizing difficulties in clinical and community samples of adolescents [7,26].

### 1.2. Maternal and Paternal Support and Youth Internalizing and Externalizing Symptoms

Identifying factors that could protect youth from the strong effects of friend internalizing and externalizing symptoms on adolescent internalizing and externalizing symptoms merits attention. So too does identifying factors that may exacerbate the effects of best friend symptoms of psychopathology. Several individual child and peer relationship factors have been previously implicated as protective (e.g., sex, self-regulation skills, friendship quality) [5,27,28] or as risk (e.g., peer rejection) [29] factors. However, research on the potential moderating effects of parental support on the relation between friend symptoms of psychopathology and youths’ psychopathology has been surprisingly limited. This may be due, in part, to long-standing, but incorrect, assumptions that peers “matter” more than parents during adolescence. Researchers have long noted that parental support continues to play a significant and unique role in fostering positive socioemotional development and well-being during adolescence [30]. For instance, numerous studies highlight the positive contributions of maternal support to slower growth of depressive symptoms during adolescence [31,32], while lower levels of maternal support have been associated with increases in youth externalizing symptoms [33]. In addition, extensive research has demonstrated that maternal support buffers the effects of stress on depressive symptoms among adolescents [10,34,35]. With relation to paternal support, however, the literature pertaining to the adolescent development of psychopathological symptoms is limited and existing findings are decidedly mixed. For example, in one study, it was reported that father support was unrelated to adolescent adjustment [36]; yet, in another investigation, paternal support was found to be negatively associated with youth depression [37].

Although limited in scope, it has been found that maternal support protects youth from the negative effects of peer contagion. For example, in a study examining the effects of peer influence on risky behavior, peer influence was greater among adolescents who reported high levels of dissatisfaction in their relationship with their mothers [38]. Similarly, researchers have found that adolescents are more likely to be affected by their peers’ substance use if they experience less maternal support [39]. This might be because youth with supportive relationships with their mothers may be able to seek help from them when they experience stress/pressure from friends, making them less likely to be overly dependent on their friends, and thus less likely to be influenced by their behavior in contrast to those who experience unsupportive relationships with their mothers [39]. Significantly, however, researchers are yet to examine maternal support in a study that also examines potential cross-over effects, and we were not able to locate a single study in which paternal support was examined as a moderator of peer contagion. Fathers, however, may also serve as protective figures given the unique features of father–child relationships (e.g., elevated levels of play, humor, and companionship) and that paternal support has been negatively associated with youth mental health issues [37].

### 1.3. Current Study

The current study addressed the limitations of the extant literature by examining the unique effects of best friend internalizing and externalizing symptoms as predictors of adolescent internalizing and externalizing symptoms, and the impact of maternal and paternal support on these relations over a two-year period (from Grade 6 (G6) to Grade 8 (G8)). Our focus on early (10–14 years) and middle (15–18 years) adolescence is significant as it is when mental health difficulties increase and when peers become increasingly influential on adjustment outcomes. As past research shows that friendship quality is related to maternal support and moderates the effects of peer contagion [28], we conservatively controlled for the effects of G6 friendship quality in our study to examine the potential unique moderating effect of maternal and paternal support.

Based on past research, we hypothesized that best friend’s internalizing (at G6) would be positively associated with youth internalizing and externalizing symptoms at G8, and best friend’s externalizing symptoms (at G6) would be positively associated with youth internalizing and externalizing symptoms at G8. If supported, both sets of findings would provide novel evidence of cross-over effects. We also predicted that maternal and paternal support would moderate the effects of best friend’s internalizing and externalizing symptoms on adolescent internalizing and externalizing symptoms, such that youth with high levels of maternal or paternal support would be less likely to be negatively impacted by their best friends’ internalizing and externalizing problems in relation to their own internalizing and externalizing problems. In light of evidence implicating adolescent sex as a moderator of peer contagion effects within friendship dyads (i.e., greater for girls than boys) [5] but mixed evidence as to whether girls and boys are similarly or differently impacted by their relationships with their mothers and fathers [8,40], we explored the impact of adolescent sex on the hypothesized effects without any a priori expectations.

## 2. Materials and Methods

### 2.1. Participants

Participants were involved in a large prospective study that examined school transitions and close relationships during late childhood and adolescence [41]. A total of 385 young adolescents (Grade 6 *M*_age_ = 13.64 years; 53% female) were selected for this study because they completed measures in both the school and laboratory portions of the project (see below) when they were in Grade 6, and two years later, when they were in Grade 8. This selected subsample was ethnically and racially diverse, with 40% being identified by their mothers as a racial or ethnic minority (15% as African American, 15% as Asian, 10% as Hispanic/Latino). In terms of parental education, 68% of the mothers (68% of the fathers) had a university degree, 21% had some college education (13% of the fathers), and 9% had high school and vocational education (12% of the fathers). Comparisons of participants who were included and those who were not on the study constructs did not reveal any significant differences (output available by request).

### 2.2. Procedures

As described previously [41], study measures were completed during school time and also at a university laboratory. Of interest in this study was one school measure (friendship nominations) completed in group-format during G6, which, along with another measure not of interest in this study (the extended class play) [41], took participants approximately one hour to complete. Teachers were not involved in data collection. Participants were assured of confidentiality, and were instructed to not discuss their answers with their classmates. Next, the G6 participants with mutual friends (~70% of the larger sample) were invited to visit the university laboratory together several months later to complete additional measures (including surveys of parental support and friendship quality) and complete activities. These participants were then invited to complete additional measures, without any friends present, two years later when they were in G8. During these visits, parents also completed several measures about their parenting and their children, such as measures pertaining to their parenting practices; of interest in the present study are parent reports of child internalizing and externalizing problems. All participants (and their parents) completed measures in separate rooms. Laboratory participants and their families were compensated with gift cards for this portion of the study; the laboratory session lasted approximately two hours.

### 2.3. Measures

#### 2.3.1. School Measure

Friendship nominations (G6). Participants wrote the names of their same-sex “very best friend” and their “second best friend” in their grades and schools [42]. Same-sex friendships were examined because the large majority of adolescents have same-sex best friends and numerous studies have indicated important differences in the function [43], quality, and influence of same-sex versus other-sex friendships during adolescence [44,45]. Only mutual (reciprocated) best friendships were subsequently considered. Consistent with procedures utilized in other studies [43], friendships were considered mutual if they were each other’s very best or second best friend choice (~70% of participants had at least one same-sex mutual friendship [41]).

#### 2.3.2. Laboratory Measures

Adolescent internalizing and externalizing symptoms (G6 and G8). To measure adolescent internalizing and externalizing symptoms, the mothers of participants (both the target participants and their best friends) completed the 120-item Child Behavior Checklist (CBCL) [46,47], which has been found to have strong psychometric properties in non-clinical samples. Items were answered on a 3-point Likert scale (0 = not true; 1 = somewhat or sometimes true; 2 = very true/often), and mean scores for the broad-band internalizing (e.g., “Fears going to school”, “Feels worthless or inferior”) and externalizing scales (e.g., “Argues a lot”, “Breaks rules at home, school or elsewhere”) were calculated (α = 0.86, 0.89, 0.84, and 0.82 for G6 internalizing and externalizing symptoms and G8 internalizing and externalizing symptoms, respectively). The CBCL is widely used internationally, including the United States (USA), and is known to have strong psychometric properties with the coefficient alphas ranging from 0.78 to 0.97 for the problem scales (0.90 for the internalizing problems subscale and 0.94 for externalizing problems subscale) demonstrating high internal consistency [47].

Maternal and paternal support (G6 and G8). The Network of Relationships Inventory (NRI) [48], a 33-item self-report measure of perceived social support, was used to assess youths’ perceptions of maternal and paternal support. Participants rated the extent to which each item, on a 5-point scale, described their relationships with their mothers and their fathers (e.g., “How often do you spend fun time with your mother/father?”, “How often do you and your mother/father disagree and quarrel with each other?”). The NRI includes several subscales. Consistent with past research [49], social support scales were created by averaging scores on seven subscales (reliable alliance, reassurance of worth, instrumental aid, companionship, affection, intimate disclosure, and nurturance) for mothers (α = 0.92) and fathers (α = 0.94) separately. The NRI has been used extensively with participants in the USA, and has demonstrated good internal consistency, with Cronbach’s alphas ranging from 0.90 to 0.91 for the parent support scales [48]. The NRI subscale on supportive relationships has been found to be negatively associated with adolescent internalizing symptoms [34,50].

Friendship quality (G6). Participants also reported on the quality of the relationship with their best friends, with whom they visited the laboratory, with the 41-item Friendship Quality Questionnaire-Revised (FQQ-R) [43]. In light of previous research showing that the quality of the friendship moderates best friend influence [28] and to evaluate the unique moderating effects of the parent–adolescent relationship, the 32-item averaged total positive friendship quality scale was utilized as a control variable in the present study (α = 0.94). The FQQ-R has been widely used to study friendship quality among adolescents in the USA. The measure has good internal consistency (coefficient alphas ranging from 0.73 to 0.90 for the different subscales) [43]. The FQQ-R has also shown strong validity; for example, the six subscale scores have been found to predict child loneliness [43].

### 2.4. Data Analytic Approach

Zero-order correlations among the study variables were first examined. Next, Mplus version 6.12 [51] was used to estimate a series of path models with full information maximum likelihood estimation with robust standard errors. Missing data were minimal, and full information maximum likelihood estimation appropriately handled missing data. The path models, as depicted in Figure 1, were identical with the exception of the moderator variable, which was either maternal or paternal support. Covariances between exogenous variables (G6 youth and best friend internalizing and externalizing problems, maternal/paternal support) were estimated, and so were the covariances between the two endogenous variables (G8 adolescent internalizing and externalizing problems). The stability paths (from Time 1 to Time 2) between youth internalizing and externalizing problems were also estimated, and interaction terms between the centered G6 best friend internalizing and externalizing variables and the G6 maternal/paternal support variables were also included (Figure 1).

Youth reports of G6 best friend support/quality was related to parental support in the descriptive analyses and thus was included as an exogenous covariate related to both maternal and paternal support in their respective models. Model fit was assessed with the chi-square goodness-of-fit statistics and the root-mean-square error of approximation (RMSEA; 0.08 or less), standardized root mean square residual (SRMR; 0.09 or less), and comparative fit index (CFI; 0.95 or greater) [52]. As reported below, all models provided a good fit to the data, and thus no post hoc model fitting was performed. Only significant paths are described below, and significant interactions were probed in Mplus in accordance with the procedures outlined by Aiken and West [53]. Multiple group path models with the full models were run to examine potential differences between boys and girls (with sex coded as 0 = boys, 1 = girls). Of note, all models were also run with a reduced dataset in which adolescents at G6 were randomly designated as either a target participant or a best friend (and not both). The results with this reduced data set were nearly identical to the reported results. See Prinstein [5] for a similar approach.

## 3. Results

### 3.1. Descriptive Statistics

Means, standard deviations, and bivariate correlations among the study variables are presented in Table 1. Of note, strong stability for the adolescent internalizing and externalizing problems variables was found, along with a strong positive correlation between adolescent internalizing and externalizing problems at both time-points. Adolescent internalizing problems at G6 were not related to their mutual G6 best friends’ internalizing and externalizing problems, but adolescent externalizing problems at G6 were related to their best friends’ internalizing and externalizing problems. Positive friendship quality at G6 was related to G6 maternal and paternal support, and G6 maternal and paternal support were moderately associated with each other.

#### 3.1.1. Moderation Analyses with Maternal Support

For the path model involving maternal support, there was good fit to the data: χ² (8) = 11.96, *p* = 0.15, RMSEA = 0.06, 90% CI [0.000, 0.127], SRMR = 0.046, CFI = 0.96 (see Figure 2 for standardized path coefficients). Significant stability was found for the adolescent internalizing and externalizing problems variables from G6 to G8. Furthermore, G6 adolescent internalizing problems predicted decreases in G8 externalizing problems, but the path from G6 adolescent externalizing problems to G8 adolescent internalizing problems was not significant. Figure 2 also shows unique and prospective associations between G6 best friend internalizing and externalizing problems and G8 adolescent externalizing problems.

Path coefficients represent the standardized results. Non-significant paths remain in the model and are displayed as dashed lines. Within-time covariances are not displayed for ease of communication, but were tested in the model and reported above.

Specifically, G6 best friend internalizing problems predicted decreases in G8 adolescent externalizing problems. In contrast, G6 best friend externalizing problems predicted increases in G8 adolescent externalizing problems. A significant interaction between G6 best friend internalizing and G6 maternal support when predicting G8 adolescent externalizing problems was also evinced. Follow-up simple slope analyses showed that G6 best friend internalizing problems (β = −1.35, *p* = 0.01) at low levels of maternal support were stronger predictors of G8 externalizing problems relative to G6 best friend internalizing problems (β = −0.85, *p* = 0.01) at high levels of maternal support.

Although not depicted in the figure, within-time positive associations were found between G6 adolescent and best friend internalizing and externalizing problems, and G8 adolescent internalizing and externalizing problems (all *p*s < 0.01). G6 maternal support was also related positively with G6 best friend quality and negatively to G6 adolescent externalizing problems (*p*s < 0.005). 

#### 3.1.2. Moderation Analyses with Paternal Support

There was also good fit of the data to the model with paternal support: χ² (8) = 9.81, *p* = 0.27, RMSEA = 0.024, 90% CI [0.000, 0.067], SRMR = 0.022, CFI = 0.992. Standardized path coefficients are presented in Figure 3. Of note, in this model, significant stability for the adolescent internalizing and externalizing variables was found. There were no other significant predictors of G8 adolescent internalizing problems, but G6 best friend internalizing problems were a significant and negative predictor of G8 adolescent externalizing problems. There was also a significant interaction effect involving G6 best friend internalizing problems and G6 paternal support. Simple slope analyses showed that G6 best friend internalizing problems were a stronger predictor of G8 adolescent externalizing problems at low (β = −1.15, *p* = 0.001) versus high (β = −0.71, *p* = 0.001) levels of paternal support. Also evident in this model, but not displayed in the figure, were significant and positive within-time associations between G6 adolescent and best friend internalizing and externalizing variables and the G8 adolescent internalizing and externalizing variables. G6 paternal support and best friend quality were also significantly and positively related, and G6 paternal support and G6 adolescent externalizing problems were significantly and negatively associated (all *p*s < 0.001).

Path coefficients represent the standardized results. Non-significant paths remain in the model and are displayed as dashed lines. Within-time covariances are not displayed for ease of communication, but were tested in the model and reported above.

#### 3.1.3. Evaluation of Sex Differences

Whether sex moderated the proposed associations was tested using a multiple group analysis in which a fully unconstrained model (all regression paths and covariances freely estimated for both sexes) was compared to a fully constrained model (all regression paths and covariances set equal for both sexes). A significant *χ*^2^ difference test between the constrained and free to vary models indicated differences across sex in the model involving maternal support, Δ*χ*^2^ (32) = 272.75, *p* < 0.001, but not paternal support, Δ*χ*^2^ (27) = 22.49, *p* > 0.05. Sequentially testing each path and covariances for sex differences revealed significant differences in a total of three within-time associations in the model. A final model with these three paths free to vary by sex demonstrated acceptable fit to the data: *χ*^2^ (39) = 38.05, *p* > 0.05, RMSEA = 0.001, 90% CI = [0.00, 0.05], SRMR = 0.08 and CFI = 1.00. G6 adolescent internalizing problems were more strongly correlated with G6 best friend externalizing problems for girls (β = 0.39, *p* = 0.001) relative to boys (β = 0.19, *p* = 0.03). G6 maternal support was associated negatively with G6 adolescent internalizing problems for boys (β =−0.25, *p* = 0.01) but not girls (β = −0.04, *p* = 0.60). Finally, G6 maternal support was associated negatively with G6 adolescent externalizing problems for girls (β = −0.31, *p* = 0.001) but not boys (β = −0.12, *p* = 0.23).

## 4. Discussion

This study examined the effects of friend internalizing and externalizing difficulties on the development of adolescent internalizing and externalizing difficulties to test for unique, and potentially, cross-over effects. We also evaluated the moderating role of parental support, with the novel consideration of both maternal and paternal support as protective factors of peer contagion. Three major findings emerged, which extend what is known about peer contagion in significant ways. First, a unique and positive effect of best friend externalizing problems on later adolescent externalizing problems was evidenced. Second, a cross-over but negative effect was found for best friend internalizing problems on later adolescent externalizing problems. Third, both maternal and paternal support emerged as moderators of the relationship between best friend internalizing problems and later adolescent externalizing difficulties, such that best friend internalizing problems were stronger predictors of later adolescent externalizing difficulties at low levels of maternal and paternal support than at high levels of maternal and paternal support. These and other findings and their implications are discussed next.

### 4.1. Unique Effects of Friend Internalizing and Externalizing Difficulties

Our longitudinal assessments of adolescents’ internalizing and externalizing difficulties allowed us to detect unique effects of best friend externalizing difficulties on the development of externalizing difficulties. Although not surprising given the strong evidence of the peer contagion of externalizing problems within adolescent friendships, the findings are notable because they are among the first to show evidence of unique effects. Thus, regardless of their levels of internalizing difficulties, adolescents’ friendships appear to uniquely foster externalizing problems, perhaps due to shared activities and the modelling of aggressive and delinquent behaviors. Similar unique effects, however, were not evinced for friend internalizing difficulties. Additional research will be needed to replicate these findings, but it may be that the peer contagion of internalizing problems reported previously could be explained, in part, by the stressors associated with having a friend with externalizing difficulties.

Notably, we did find evidence of a cross-over effect, although it was in the negative direction and only involved friend internalizing, and not friend externalizing, problems. More specifically, G6 friend internalizing problems negatively predicted G8 adolescent externalizing problems. This finding is novel and suggests that having a best friend with internalizing problems may not always have negative outcomes, as typically reported and discussed in the extant literature [54]. Instead, it may actually confer some benefits for youth with externalizing problems. The specific processes involved will need to be investigated, but we speculate that having a friend with internalizing problems may deter adolescents from engaging in externalizing problems as externalizing behaviors oftentimes involve risk taking about which those suffering from internalizing problems would be fearful and anxious. Co-rumination processes may also help to discourage externalizing behaviors by perhaps changing opinions about the behaviors or displacing time that would be spent engaging in externalizing behaviors. The protective effects may also be due to the contagion of social problem-solving and coping strategies, as inhibited and anxious children and adolescents tend to select avoidant solutions for dealing with stressors, such as withdrawing from situations when faced with social conflicts [55,56]. While the heavy reliance on avoidant strategies is generally considered maladaptive for anxious youth, it is possible that youth with externalizing tendencies who observe these strategies may learn to employ these non-confrontational strategies as part of their larger social problem-solving repertoire and thereby decrease their use of aggressive behaviors when facing interpersonal conflicts.

### 4.2. Moderating Role of Parent Support

In support of our hypotheses, we also found that maternal and paternal support both moderated the effects of G6 best friend internalizing problems on G8 youth externalizing problems, such that the cross-over associations between friend internalizing and adolescent externalizing problems were stronger at *low* maternal or paternal support. However, because the association between G6 best friend internalizing problems and G8 adolescent externalizing problems was unexpectedly negative, this finding suggests that high parent support weakened the potentially beneficial association between best friend internalizing difficulties and lower levels of adolescent externalizing difficulties. This pattern differs from the findings of previous studies examining parent support as a moderator of peer contagion, since those studies showed that high parent support served as a protective factor that weakened the association between friend externalizing behaviors and adolescent externalizing behaviors [38,39], which was not found in the present study. However, the current study is the first to evaluate cross-over effects, and the findings do align with those from the previous studies as the effects of friend externalizing behaviors appeared to be strongest at low (versus high) levels of parental support. This may be because adolescents receiving little support for their parents may instead turn to their friends for support. As a result, any friend influence may be amplified when parental support is low. Thus, although initially puzzling, we think that our findings suggest that high levels of parent support may not only protect against peer contagion of negative behaviors (as found in other research), but perhaps also diminish potentially helpful influences from friends (as found herein).

### 4.3. The Role of Sex

No significant sex differences were found for the predictive associations between G6 best friend internalizing and externalizing symptoms and G8 adolescent internalizing and externalizing symptoms. The few sex differences that emerged involved concurrent associations in G6, with maternal support negatively related to internalizing problems only for boys, and externalizing problems only for girls. Additionally, adolescent internalizing problems and best friend externalizing problems in G6 were more highly correlated among girls compared to boys. Taken together, our findings suggest that associations between the internalizing and externalizing symptoms of adolescents and their best friends were similar across sexes. Several prior studies of associations between social relationships and internalizing and externalizing symptoms have similarly found no sex differences [8,57]. However, other research has shown that sex may alter peer contagion effects; for example, being exposed to peers’ externalizing problems predicted similar problems for girls, but not boys [58]. Similarly, friends’ self-injurious behavior predicted adolescents’ own non suicidal self-injurious behavior just for girls, but not for boys [59]. Therefore, further research will be needed to clarify the mixed findings in this area of research and when and why peer contagion effects differ for boys and girls.

### 4.4. Clinical Implications

If replicated, our findings may have several important implications for clinical psychologists, school psychologists, and other mental health professionals working with youth. Best friend externalizing problems were associated uniquely with increases in youth externalizing problems, and best friend internalizing problems were associated uniquely with decreases in youth externalizing problems, suggesting that practitioners working with youth with externalizing problems should more carefully consider the role of peers and their potential unique contribution to maintaining, increasing, or decreasing externalizing behaviors. Assessing peer mental health and behavioral issues may not only help obtain a fuller understanding of the youth’s existing issues, but may also help to inform treatment options. For example, if it is established that youth may be exhibiting externalizing problems due to peer influence, they may be encouraged to form relationships with peers without externalizing difficulties, or even specifically assigned to spend time in the company of other peers without externalizing problems. Indeed, past research has shown support for the idea, where youth aggressive behavior decreased when they were paired with a non-aggressive peer [60].

Our findings also highlight the importance of assessing parenting-related factors while working with youth with mental health and behavior concerns. Both maternal and paternal support emerged as moderators of the association between best friend internalizing problems and adolescent externalizing problems, illustrating the complex interplay between parent and peer relationships in the development of adolescent problem behaviors and suggesting that it is likely beneficial to include parents in the treatment process. Indeed, interventions for children with mental health and behavioral concerns often include parents (e.g., parent–child interaction therapy) [61], and have been found to be efficacious in reducing child mental health and behavioral problems [62].

### 4.5. Limitations and Future Directions

The findings of the current study should be considered within the context of its limitations. First, we used questionnaires only to assess our constructs, giving rise to the possibility of mono-method bias. On a related note, we used parent reports of child internalizing and externalizing behaviors, which may raise concerns about biased reporting. Indeed, research suggests that there are often discrepancies between parent reports and youth reports of youth mental health problems [63]. Thus, the findings from the current study can be corroborated in future studies by including different forms of assessments such as observations, and by including different informants for sources of data. Second, we used a community sample in our study, and thus the findings may not generalize to clinical samples. Third, in order to measure peer influence, we only focused on the effects of the best friend’s influence. While best friendships are an important peer relationship and therefore a key potential source of peer influence, it is likely that youth are also affected by other peers, including other friends and acquaintances in the larger peer group. Thus, future research may benefit by extending the findings from this study by considering the role of peer groups and peer networks.

## 5. Conclusions

Notwithstanding our limitations, our study makes several important contributions. It is the first to examine the unique effects of best friend internalizing and externalizing problems on youth internalizing and externalizing problems, and the moderating role of both maternal and paternal support in a sample of young adolescents as they transition into middle adolescence, a population highly vulnerable to experiencing mental health issues. The longitudinal design of the study allowed us to examine the relations between the constructs over time in a large and relatively diverse sample. The findings indicate that best friend internalizing problems predict decreases in youth externalizing problems, while best friend externalizing problems predict increases in youth externalizing problems over time. In addition, the findings highlight the role of parental support: best friend internalizing problems were stronger predictors of later adolescent externalizing difficulties at low levels of both maternal and paternal support than at high levels of maternal and paternal support. Thus, our findings shed light on the importance of considering the impact of peer influences as well as parental influences while working with youth with behavioral and mental health concerns.

## Figures and Tables

**Figure 1 children-08-00306-f001:**
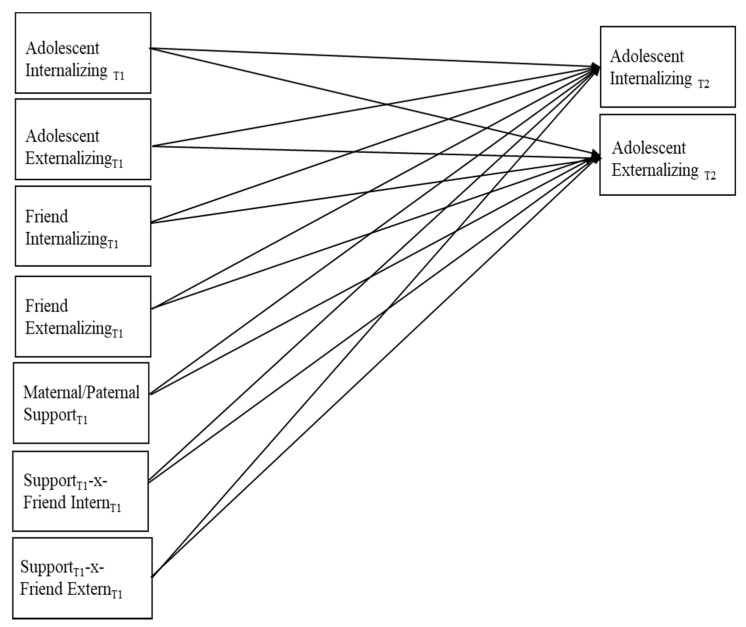
Hypothesized path model. Within-time covariances are not displayed for ease of communication.

**Figure 2 children-08-00306-f002:**
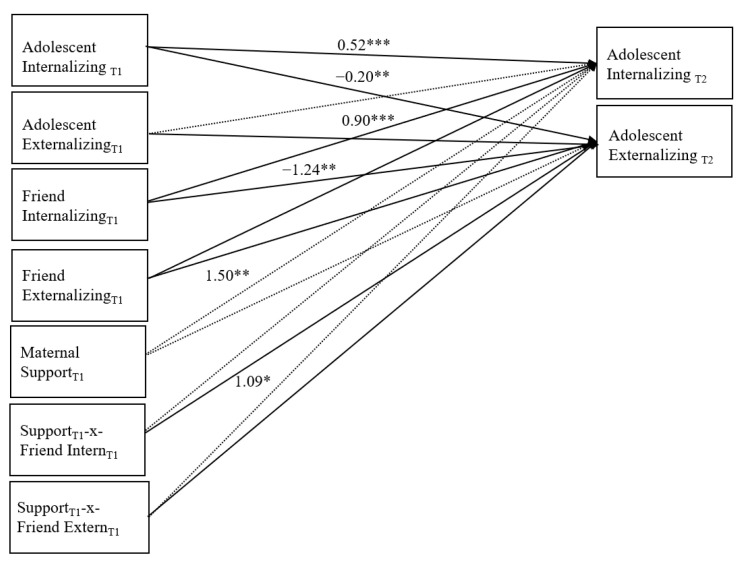
Final path model with maternal support.* *p* < 0.05, ** *p* < 0.01, *** *p* < 0.001

**Figure 3 children-08-00306-f003:**
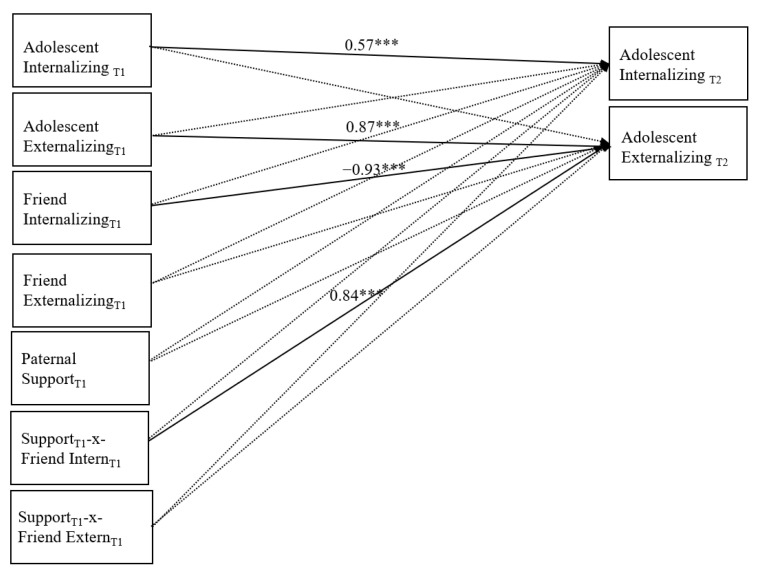
Final path model with paternal support. *** *p* < 0.001.

**Table 1 children-08-00306-t001:** Summary of intercorrelations, means and standard deviations on key study variables.

		1	2	3	4	5	6	7	8	9	10
1.	G6 internalizing	−									
2.	G6 externalizing	0.67 **	−								
3.	G8 internalizing	0.55 **	0.36 **	−							
4.	G8 externalizing	0.32 **	0.69 **	0.58 **	−						
5.	BF G6 internalizing	0.15 *	0.25 **	0.01	0.07	−					
6.	BF G6 externalizing	0.25 **	0.27 **	0.15	0.14	0.68 **	−				
7.	G6 Mother Support	−0.08	−0.19 **	−0.06	−0.17	−0.11	−0.12	−			
8.	G6 Father Support	−0.05	−0.18 **	−0.04	−0.18 *	−0.10	−0.09	0.54 **	−		
9.	G6 Friendship quality	−0.08	−0.06	0.07	0.02	−0.13 *	−0.07	0.18 **	0.19 **	−	
10.	Sex (female: N = 205; % = 53.2)	0.01	−0.13 *	0.18 *	−0.06	−0.00	−0.13 *	0.08	−0.02	0.22 **	−
M		6.23	6.13	5.39	4.63	6.25	6.08	4.16	3.94	3.96	0.53
SD		6.04	6.19	4.95	5.02	6.05	6.21	0.56	0.68	0.60	0.50

* *p* < 0.05, ** *p* < 0.01.; BF = Best Friend; SD = Standard Deviation.

## Data Availability

Data presented in this study are available on request from author Rubin. Data are not publicly available in accord with consent provided by participants on the use of confidential data.

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
