# Peer review of "Peer Influence during Adolescence: The Moderating Role of Parental Support"

_children, 2021, doi:10.3390/children8040306_

Round 1

Reviewer 1 Report

Recommendation: Revise and Resubmit    The manuscript addresses a topic that is consistent with the scope and aims of this Journal   It contributes new knowledge to the specialty in an important issue for adolescent health. Writing style, organization, and clarity is adequate.    The study shows new findings.   The use of language is clear and precise.   The ideas are presented in an economical way.   The content is interesting.   The content is well organized with logical flow.    The work is grounded in recently published literature.   The purpose of the manuscript is very important and innovative for adolescent health: They considered both internalizing and externalizing difficulties in the same study with adolescents, and the contributions of parents (for instance, the potential moderating effects of parental support on the relation between friend symptoms and youths’ internalizing and externalizing problems) and the best friend (via increased general stress or the peer contagion processes of mental health difficulties and psychopathology: adolescents appear to be at increased risk for the development of adjustment difficulties,  depression or anxiety symptoms when their friends are experiencing difficulties themselves; and similar contagion effects in a variety of externalizing behaviors, including relational and instrumental aggression, alcohol and drug use, risky sexual behavior, and suicidal behavior). They examined the unique effects of best friend internalizing and externalizing symptoms as risk factors, and the protective role of maternal and paternal support, in the development of adolescent internalizing and externalizing symptoms.  

The sample appears to be adequate with appropriate data:

Using a sample of 385 adolescents, in a longitudinal study. They examined the unique effects of best friend internalizing and externalizing symptoms as predictors of adolescent internalizing and externalizing symptoms, and the impact of maternal and paternal support on these relations over a two-year period (from grade 6 (G6) to grade 8 (G8), during late childhood and adolescence).

The procedure and instruments used are appropriate to the sample and to the proposed variables and objectives, however the specific psychometric properties of them on the country where the adolescents are from should be mentioned in the text, as well as the reliability found in these instruments in this study sample.

The design and statistical methods are adequate: Mplus version 6.12 was used to estimate a series of path models with maximum  likelihood estimation with robust standard errors. Models were identical with the exception of the moderator variable, which was either maternal or paternal support. Covariances between exogenous variables (G6 adolescent and best friend internalizing and externalizing problems, maternal/paternal support) were estimated, and so were the covariances between these two endogenous variables (G8 adolescent internalizing and externalizing problems). Model fit was assessed with the chi-square goodness-of-fit statistics and the root-mean-square error of approximation (RMSEA; .08 or less), standardized root mean square residual (SRMR; .09 or less), and comparative fit index (CFI; .95 or greater). All models provided a good fit to the data, and significant interactions were probed in Mplus. Multiple group path models with the full models were run to examine potential differences between boys and girls

  They hypothesized that best friend’s internalizing symptoms (at G6) were positively associated with youth internalizing and externalizing symptoms at G8, and best friend’s externalizing symptoms (at G6) were positively associated with youth internalizing and externalizing symptoms at G8. They also predicted that maternal and paternal support moderate the effects of best friend’s internalizing and externalizing symptoms on adolescent internalizing and externalizing symptoms such that youth with high levels of maternal or paternal support would be less likely to be negatively impacted by their best friends’ internalizing and externalizing problems in relation to their own internalizing and externalizing problems.   The final results and conclusions of this study help control for interrelationships and identify key predictors of adolescent health. After carrying out the statistical analysis, they found: a unique and positive effect of best friend externalizing problems on later adolescent externalizing problems; a cross-over, but negative effect, for best friend internalizing problems on later adolescent externalizing problems; both maternal and paternal support emerged as moderators of the relation between best friend internalizing problems and later adolescent externalizing difficulties, such that best friend internalizing problems were stronger predictors of later adolescent externalizing difficulties at low levels of maternal and paternal support than at high levels of maternal and paternal support (best friend internalizing and externalizing problems predict adolescent externalizing problems, particularly when maternal or paternal support is low). The findings suggest that associations between the internalizing and externalizing symptoms of adolescents and their best friends were similar across sexes. These findings provide novel evidence of cross-over effects As they commented, in future research, it is necessary to extending the findings from this study by considering the role of peer groups and peer networks, not only the best friends. Also, it is necessary to include in future studies also the own report of adolescents of his/her youth mental health problems apart from the parent-report of child internalizing and externalizing behaviors, and compare these results with other that could come from Clinical adolescent samples The differences found between their predictions and the results of their study are justified with good arguments in their conclusions, trying to explain the discrepancies. They also adequately refer in the discussion their results in relation to those existing in previous research. They also propose clinical actions to be taken based on their results.   These results could improve knowledge about psychological health during adolescent caregiving, and this will help prevent and treat earlier, effectively and adequately to this population in conditions of great vulnerability and risk in their physical and psychological health.

Reviewer 2 Report

This study analysed the moderating role of parental support in the relations between peer mental health and adolescents’ internalising and externalising difficulties in a sample of 385 youths aged 13-14 at the first measurement occasion. Results indicated that internalising difficulties in adolescents’ best friends predicted subsequent reductions in externalising behaviours. This association was moderated by both maternal and paternal support with lower levels of parental support resulting in a stronger negative effect of best fried internalising problems on later adolescent externalising problems. Best friends’ externalising behaviours, on the other hand, predicted increases in adolescents’ externalising difficulties over time. This study presents interesting and novel findings in the relations between peer mental health, parental support and adolescents’ development of internalising and externalising problems. See my comments below:

  1. Typo in Page 2, line 75: “in the development pf adolescent internalizing”
  2. Please add some more details on other measures administered as part of the current study (Page 5).
  3. Please add details on the age of adolescents at the second measurement occasion. (Page 5).
  4. Please add a justification for why only same-sex friendships were considered. (Page 5)
  5. I am missing a statistical analysis section in the methods part. Considering that the presented analyses are likely not familiar to all readers, a brief introduction would help less statistically versed readers follow the results section better. (Page 4-5)
  6. Please add somewhere how missing data was handled. Or was there none?
  7. Table 1, please present N and % for gender rather than mean and SD. Are these correlations Pearson’s correlation coefficients? This wouldn’t be appropriate for gender, please use an appropriate method here, e.g., point-biserial correlation coefficient and clarify what method was used.
  8. Figure 2 is not clear: some paths are shown as significant even though they are not, e.g., best friend externalising to internalising. Please correct this.
  9. Page 10, line 405: When discussing the unique effects of internalising and externalising you state that “even when adolescents are also suffering from internalizing problems, their friendships appear to uniquely foster externalizing problems”. This statement is not correct since having a score on internalising does not mean that an individual “suffers” from internalising problems. Please reformulate this to be clear that you are controlling for the effect of internalising when investigating the effect of externalising behaviours.
  10. While I agree that your findings are in line with previous studies in that low parental support leads to stronger effects of externalising on adolescent mental health, I think the explanations given for this finding contradict your results. For example, shouldn’t better maternal support e.g. to seek maternal help when stressed, lead to a further reduction in internalising scores, that is strengthen rather than weaken the negative association with friend externalising problems? Please expand this section since this is one of the most interesting findings of this study. (Page 10-11)
  11. The section on the role of sex could be formed out a bit more. You say that further research needs to clarify the mixed findings in this area of research but you only cite studies that, like the current study, did not find any sex differences. Please connect your findings a bit more to the literature on sex differences in peer effects on adolescent mental health. (Page 11)
  12. In the conclusion, you state that “The findings suggest that best friend internalizing and externalizing problems predict adolescent externalizing problems, particularly when maternal or paternal support is low". If I understand correctly, parental support only moderated the relation between internalising and subsequent externalising behaviours. Please reformulate this sentence as it is currently misleading.
